# Shotgun metagenomics captures more microbial diversity than targeted 16S rRNA gene sequencing for field specimens and preserved museum specimens

**Joseph D. Madison** *, **Brandon C. LaBumbard, Douglas C. Woodhams**

Department of Biology, University of Massachusetts Boston, Boston, Massachusetts, United States of America

* josephdmadison7@gmail.com

**Data Availability Statement:** The workflow used for processing and analysis within Qiita is available as a study: 'Shotgun versus 16S - Museum and Fresh Leopard Frog Guts - ID 14549'. Additional

## Abstract

The use of museum specimens for research in microbial evolutionary ecology remains an under-utilized investigative dimension with important potential. Despite this potential, there remain barriers in methodology and analysis to the wide-spread adoption of museum specimens for such studies. Here, we hypothesized that there would be significant differences in taxonomic prediction and related diversity among sample type (museum or fresh) and sequencing strategy (medium-depth shotgun metagenomic or 16S rRNA gene). We found dramatically higher predicted diversity from shotgun metagenomics when compared to 16S rRNA gene sequencing in museum and fresh samples, with this differential being larger in museum specimens. Broadly confirming these hypotheses, the highest diversity found in fresh samples was with shotgun sequencing using the Rep200 reference inclusive of viruses and microeukaryotes, followed by the WoL reference database. In museum-specimens, community diversity metrics also differed significantly between sequencing strategies, with the alpha-diversity ACE differential being significantly greater than the same comparisons made for fresh specimens. Beta diversity results were more variable, with significance dependent on reference databases used. Taken together, these findings demonstrate important differences in diversity results and prompt important considerations for future experiments and downstream analyses aiming to incorporate microbiome datasets from museum specimens.

## Introduction

The use of nucleic acid sequencing for analyzing museum specimens has great potential given the emergence of high-throughput molecular technologies [1]. These technologies applied to museum specimens can be used to ask questions of temporal relevance to ecology and evolution not otherwise accessible to experimental inquiry. Unfortunately, the conditions of specimen preservation techniques such as formalin fixation and storage in alcohol fluids (e.g.

code used for nucleic acid data processing and related statistical analyses are available in a GitHub repository at https://github.com/kvasir7/16S_shotgun_comparison_project. Raw sequencing files in fastq.gz format can be found on the Sequence Read Archive as BioProject PRJNA836960.

**Funding:** JDM is supported by the NSF Postdoctoral Research Fellowships in Biology Program (Award 1907311; www.nsf.gov) and DCW by NSF Career and Infrastructure Awards (Award IOS-845634 and BII 2120084; www.nsf.gov). Any opinions, findings, and conclusions or recommendations expressed in this material are those of the authors and do not necessarily reflect the views of NSF. NSF had no role in study design, data collection and analysis, decision to publish, or preparation of the manuscript.

**Competing interests:** The authors have declared that no competing interests exist.

ethanol, isopropanol) have been a major hindrance to the use and full exploitation of these molecular technologies. This has resulted in a concerted effort to improve both wet-lab and downstream analytical methods for overcoming these challenges such that molecular technologies might be better used in conjunction with the vast catalogs of biological collections represented in museum repositories [2–4].

One area of recent methodological focus for molecular analysis of preserved specimens is the specimen-associated microbial community or microbiome. Characterizing the microbiome of fluid-preserved museum specimens is a rapidly increasing area of focus due to its relevance in biology broadly, but especially in the disciplines of ecology and evolution [5,6]. The most common way to characterize host-associated microbiomes is through nucleic acid isolation followed by 16S rRNA gene sequencing. However, there are limitations to 16S rRNA gene sequencing including irrelevance to host-associated taxonomic groups with differing marker genes (e.g. ITS in fungi) [7], groups with no universal marker genes (e.g. viruses) [8] and bias in 16S rRNA gene copy number among different bacteria that is not easily corrected [9]. Alternatively, the use of shotgun metagenomic sequencing and associated analytical tools that generate large numbers of sequence reads might allow higher resolution analysis. Where long-read sequencing might otherwise be used, high-throughput short-read shotgun metagenomics is useful in dealing with the highly degraded (i.e. short reads) and low-input characteristics of nucleic acids associated with museum specimens. Here, we compare both 16S rRNA gene sequencing and short-read shotgun metagenomics methods for examining museum specimen-associated gut microbiomes.

While there has been previous work demonstrating the generally higher resolution of shotgun sequencing among these two sequencing technologies for microbial applications in fresh specimens [10–13], there has been no previous work that we are aware of examining taxonomic and diversity representation of these two sequencing methods for museum specimen-derived microbiota. In this study, we therefore hypothesized that previous results from fresh specimens would extend to museum specimens, with differences in taxonomic predictions between sequencing technologies and related diversity analysis of gut-isolated microbial nucleic acids of fluid-preserved specimens of Northern leopard frogs, *Rana pipiens*. Additionally, we hypothesized that the differences in museum specimen-derived results would diverge from the differences observed in freshly sampled specimens. Generally confirming these hypotheses, our corresponding results indicate differences in the sequencing approaches (16S rRNA gene vs shotgun metagenomics) and specimen types (museum vs. fresh) that should be considered in future study design and experimental contexts.

## Materials and methods

### Specimen origin and storage

All museum specimens used in this study (n = 13) were originally sourced from various locales in the Midwestern States of Wisconsin and Illinois, and obtained from the Milwaukee Public Museum or the Illinois Natural History Survey/University of Illinois Natural History Museum. The specimens used were originally field collected over a 120 year period with approximately one specimen per decade (1892–2012; see supplemental specimen metadata) to capture temporal degradation effects and likely differing storage conditions common among many older museum specimens with uncertain storage histories. The fresh specimens (n = 5) were from *R. pipiens* collected in eastern South Dakota, an area that encompasses the historic midwestern Tallgrass Prairie—a biome that extends through the aforementioned Wisconsin and Illinois regions [14] and has historically similar soil microbial communities [15]. Fresh specimens were euthanized via topical application of 20% benzocaine. Specimens were collected under a

South Dakota Game, Fish and Parks Scientific Collector's Permit (#2020–2) and IACUC protocol (AUP 18–28 issued to collaborator Drew R. Davis at The University of Texas Rio Grande Valley, who did the collecting).

All collection locales over both current and historical time periods captured in this study exhibit or have exhibited similar climatic patterns and land-use dominated by mono-crop agricultural operations. All museum specimens were received in ethanol and were stored on site at the University of Massachusetts Boston in the same ethanol concentrations as received, within individual sterilized mason jars (Ball). Jar size used was variable and dependent on specimen size. After initial receipt, specimens were kept in their respective jars until dissection.

## Specimen dissections and swabbing

All dissections and swabbing were completed under aseptic conditions to include PPE and sterile equipment, where applicable. For the dissections, museum specimens were placed on 20.3 cm x 25.4 cm ABD sterile pads (McKesson Medical-Surgical Inc., Richmond, VA) within an aseptic Class II Type A2 Biological Safety Cabinet (Labgard ES, Energy Saver, Nuaire, Plymouth, MN). A 2 cm incision was then made in the ventral abdomen to expose the intestines (disposable gamma irradiated No. 12 scalpels; Swann-Morton, Sheffield, England, UK). After pinning open the body cavity, a 1 cm longitudinal incision was made along the mid-intestine. The inside of the intestine was then swabbed (Rayon bud; Medical Wire Equipment, MW113, Corsham, Wiltshire, England, UK) by twisting the swab three times (1080 degrees). Specimens from fresh caught frogs were also obtained following the protocol above using aseptic technique, with the exception that dissections were carried out in the field immediately following euthanasia, as opposed to dissection in a biosafety cabinet. The swab was next immediately placed in a micro-centrifuge tube for downstream processing (modified from Hykin [16]) to include DNA extraction and sequencing.

## Phenol-chloroform DNA extractions

Nucleic acid extractions were completed using a modified phenol-chloroform extraction for museum samples as per Campos & Gilbert [17]. Briefly, swabs were incubated in 500 μL of alkaline digestion buffer for 40 min at 100°C on a heating block (Drybath Standard 2-block, Thermo Scientific). The alkaline buffer was composed of Sodium dodecyl sulfate (BioXtra $\geq$ 99.0%, Sigma-Aldrich, St. Louis, MO) and Sodiumhydroxide solution from a 10 M stock aq. (BioUltra, Sigma-Aldrich). After cooling to room-temperature, 500 μL of phenol-chloroform-isoamyl alcohol mixture solution was added (25:24:1; BioUltra, Sigma-Aldrich). The solution was then agitated on a shaker table (Lab-line Maxirotator, Lab-Line Instruments Incorporated, Melrose Park, IL) for 5 min on the high setting. The solution was then centrifuged for 5 min at 12,000 x g (~rcf) (Centrifuge 5425, Eppendorf, Hamburg, Germany). After centrifugation, the top aqueous layer was added to 500 μL of chloroform ($\geq$ 99.5% containing 100–200 ppm amylenes as stabilizer; Sigma-Aldrich) in a new 1.5 mL microcentrifuge tube. This was followed by another round of centrifugation for 5 min at 12,000 x g (~rcf). The top layer was then added to a new 1.5 mL centrifuge tube, followed by addition of 300 μL (0.7 volume) 2-propanol (BioReagent for molecular biology $\geq$99.5%, Sigma-Aldrich) and 50 μL (~0.1 volume) of molecular biology grade 3 M sodium acetate at pH 5.2 (EMD Millipore Corp., Billerica, MA). This mixture was then centrifuged for 30 min at 12,000 x g (~rcf). Following the 30 min centrifugation, the liquid was decanted and disposed. The pellet was then washed by adding 500 μL of ethanol (85% aq.; 200 proof molecular biology grade ethyl alcohol, Sigma-Aldrich, diluted with UltraPure distilled water, (Invitrogen, Grand Island, NY), inverting once, and centrifugation for 5 min at 12,000 x g (rcf). After this centrifugation, the ethanol

solution was decanted and disposed, with any remaining solution removed with a small diameter pipette. The tubes were then briefly incubated at 70°C to remove any residual ethanol. Lastly, the pellet was re-suspended by adding 50 μL of 1X Tris-EDTA buffer solution at pH 8.0 (BioUltra, Sigma-Aldrich) and mixing with a pipette. The suspended nucleic acids were then stored in a 4°C refrigerator or -80°C freezer depending on the immediacy of downstream applications.

### Shotgun metagenomics sequencing and analysis

Library preparation for sequencing reactions was carried out using an NEB workflow for the NEBNext Ultra II DNA library prep kit for Illumina (New England Biolabs, Ipswich, MA). Paired-end 151 bp sequencing was carried out by the University of Minnesota Genomics Core on an Illumina NovaSeq 6000 using an S4 flowcell, with an average read depth of 23,844,087 reads/sample (range: 9,108,413–54,875,278/sample; see S1 Table).

For analysis, raw Illumina output was first sorted by index-linked sample ID. Sorted reads were then quality inspected with FastQC followed by host and background nucleic acid removal. This was completed by construction of custom host databases of *R. temporaria* and *B. Bufo* with Kraken2 [18]. Non-matching reads to the host database meeting the 0.5 confidence threshold were then sorted into a separate file for downstream analysis. The *R. temporaria* genome was used as a completed whole-genome reference is not available for *R. pipiens* (the host from which swabs were derived). The *R. temporaria* genome has high homology to *R. pipiens*, and the same karyotype [19], making it an acceptable reference for this filtering step. Likewise, sequences with 0.5 confidence homology to *Bufo bufo* were matched and removed as separate *Anaxyrus americanus* museum specimens were present and processed in the laboratory space used.

Background swabs processed in parallel but without specimen swabbing were also used as controls for background contamination removal. Following the same procedure as the host removal, a database was constructed of the control swabs in Kraken2 and matches were removed. A more conservative confidence threshold of 0.8 was used so as to ameliorate concerns of erroneous removal of close matches between bacteria in the sample and background (S1 Fig). Exact matches resulting in sample removal were unlikely based on differential degradation patterns in the specimen and also visual inspection of sequence taxonomy tables. After quality inspection, host removal, and background removal, the output was analyzed in the Qiita metagenomic analysis pipeline [20] and is available in the Data Accessibility and Benefit-Sharing section below.

Qiita analysis followed the recommended shotgun metagenomics pipeline. This workflow includes an initial sample adapter removal step with fastp [21], human host filtering with minimap2 (to account for sample contamination during handling [22], and taxonomic profiling via bowtie2 [23] with either the WoL reference database [24] (representing Bacteria and Archaea) or the Rep 200 database which is composed of RefSeq assemblies [25] (representing Archaea, Bacteria, Fungi, Protozoa, and Viruses), and species-level feature-table files in.qza format generated with Woltka [26]. These feature tables were then analyzed with QIIME2 [27]. Shotgun metagenomic data was not subject to rarefaction prior to downstream diversity analyses due to qualitatively increasing alpha diversity with increasing sequence depth (no plateau), and also greater uncertainty as to the effects of rarefaction on analysis of shotgun metagenomic sequencing data (as opposed to 16S rRNA gene sequencing).

### 16S rRNA gene sequencing and analysis

A subset of the same extracted samples used in shotgun metagenomic sequencing was also used for 16S rRNA gene sequencing. PCR was conducted in duplicate to amplify the V4 region

of the 16S rRNA bacterial gene (515F and 806R primers) following the Earth Microbiome protocol [28]. Following PCR, sample amplicons were pooled and then purified and normalized using a Mag-Bind EquiPure Library Normalization Kit (Omega Bio-tek, Inc., Norcross, GA, USA). The library consisted of 10 μL of each normalized sample pooled together, which was then sequenced on an Illumina MiSeq v2 300 cycles cartridge for single-read sequencing (300 bp/read).

The raw Illumina 16S rRNA amplicon data averaging 21,535 reads/sample (range: 8,263–33,259; see S2 Table) was processed and quality filtered using QIIME 2 v2020.2 [27] and classified into amplicon sequence variants (ASVs) using the DADA2 workflow [29]. Within DADA2, reads were trimmed to 150 bp based on quality score checks for all samples (S2 Fig), and bacterial taxonomy was assigned using 16S Greengenes 13_8 99% OTUs [30] reference classifier (Bacteria and Archaea). ASV reads found in extraction and PCR negative controls with more than 20 reads were deemed as contaminants and filtered out of all samples. We also filtered out reads assigned as "mitochondria" and "chloroplast". Next the dataset was rarefied at 3000 to normalize read counts across samples (3000 is based on a Qiita-derived alpha diversity rarefaction plot; S3 Fig). We subsequently generated several metrics on the dataset to analyze differences in alpha (ACE and Shannon's) and beta (Jaccard) diversity. These microbiome diversity metrics and related taxonomic summaries were determined using QIIME2 for downstream statistical analyses. For visualization, we used both the EMPeror visualization tool and innate R plotting for illustrating trends, and to describe statistical differences among methods as given below.

## Statistical analyses

All statistical analyses to include alpha and beta diversity comparisons were performed either within Qiita/QIIME2 or in R [31]. Alpha diversity was examined between shotgun metagenomic and targeted 16S rRNA gene sequencing methods, and between museum-preserved and fresh samples. Two alpha diversity indices were used in these comparisons: Shannon's diversity index [32] and the abundance-based coverage estimator (ACE) metric. Shannon diversity was used as it is a commonly used diversity metric and thus broadly interpretable. ACE was developed as an improvement on the Chao1 index [33] and is used due to its ability to correct for low represented community members and thus underrepresented abundances; a phenomena we expected from using degraded nucleic acids from preserved museum specimens.

Beta diversity and related matrix comparisons were also performed between both sequence methods (shotgun metagenomic or 16S rRNA gene) and sampling methods (museum or fresh collected). Full mantel tests and procrustes analysis on the derived Jaccard distance matrices and related principal coordinate analyses (PCOA) were used to test for matrix distance similarity and microbial community structure, respectively. Mantel tests were performed to test the null hypothesis that there is a lack of relationship between values in pairwise dissimilarity matrix comparisons. This technical question arises from the methodological use of distance matrices and their derivation method (in this case sequencing strategies), which is different from the rationale criticized by Legendre et al. [34] in landscape ecology studies.

We also qualitatively examined taxonomic differences between sequencing methods and sample types (museum or fresh). Taxonomy plots of fresh and specimen derived samples were made for 16S rRNA gene sequencing derived ASVs and of shotgun metagenomic sequencing derived species-level taxa generated from alignment-based matching with Rep 200 or WoL. Heatmaps were also made of the top-ten most represented phyla and genera among the three sequencing comparisons made (with fresh and museum combined, grouped by either phyla or genera as seen in the corresponding figures).

**Table 1. A–B. ACE comparisons between the six reference processing types using Kruskal-Wallis rank sum test followed by Dunn's post-hoc pairwise testing (Benjamini-Hochberg corrected).**

**A. Kruskal-Wallis Rank Sum Test**

| Comparison | Kruskal-Wallis chi-squared | df | P-value |
|---|---|---|---|
| Museum specimens | 30.917 | 2 | 1.93e-07 |
| Fresh specimens | 11.324 | 2 | 0.00348 |

**B. Dunn's Post-hoc Pairwise Test**

| Comparison | Z | P-unadjusted | P-adjusted (BH) |
|---|---|---|---|
| Rep200–Greengenes (Fresh Specimens) | -3.307 | 0.000943 | 0.00283 |
| WoL–Greengenes (Fresh Specimens) | -2.058 | 0.0396 | 0.0594 |
| Rep200 –WoL (Fresh Specimens) | 1.196 | 0.232 | 0.232 |
| Rep200–Greengenes (Museum Specimens) | -5.559 | 2.709e-08 | 8.126e-08 |
| WoL–Greengenes (Museum Specimens) | -2.983 | 2.856e-03 | 4.284e-03 |
| Rep200 –WoL (Museum Specimens) | 2.630 | 8.549e-03 | 8.549e-03 |

## Ethics statement

Fresh swabs used in this study were taken from voucher museum specimens collected by DRD under a South Dakota Game, Fish and Parks Scientific Collector's Permit (#2020–2). Vouchering was completed using standard best practices under an approved IACUC protocol (AUP 18–28 issued to collaborator Drew R. Davis at The University of Texas Rio Grande Valley, who did the collecting).

## Results

Community alpha and beta diversity metrics were examined for comparing shotgun metagenomic (WoL and Rep200 reference databases) and 16S rRNA gene sequencing methods (Greengenes). These metrics indicated significant differences in the predictions of shotgun metagenomic and 16S rRNA gene sequencing methods, with more limited differences observed between the different shotgun metagenomic sequencing reference databases used.

### Alpha diversity analysis

The omnibus Kruskal-Wallis tests for both alpha diversity metrics (Shannon's diversity index and ACE) indicated significant differences among shotgun metagenomic sequencing data classified with WoL, shotgun metagenomic sequencing classified with rep200, and 16S rRNA gene sequencing data classified with Greengenes ($p<0.05$; $\alpha = 0.05$; Tables 1A and 2A). Posthoc

**Table 2. A–B. Shannon's diversity index comparisons between the six reference processing types using Kruskal-Wallis rank sum test followed by Dunn's post-hoc pairwise testing (Benjamini-Hochberg corrected).**

**A. Kruskal-Wallis Rank Sum Test**

| Comparison | Kruskal-Wallis chi-squared | df | P-value |
|---|---|---|---|
| Museum Specimens | 27.185 | 2 | 1.25e-06 |
| Fresh Specimens | 9.38 | 2 | 0.00919 |

**B. Dunns Post-hoc Pairwise Test**

| Comparison | Z | P-unadjusted | P-adjusted (BH) |
|---|---|---|---|
| Rep200 –Greengenes (Fresh Specimens) | -2.616 | 0.00889 | 0.0133 |
| WoL–Greengenes (Fresh Specimens) | 0.0707 | 0.944 | 0.944 |
| Rep200 –WoL (Fresh Specimens) | 2.687 | 0.00721 | 0.0216 |
| Rep200–Greengenes (Museum Specimens) | -5.2118 | 1.871e-07 | 5.612e-07 |
| WoL–Greengenes (Museum Specimens) | -2.735 | 6.240e-03 | 9.360e-03 |
| Rep200 –WoL (Museum Specimens) | 2.477 | 1.325e-02 | 1.325e-02 |

Dunn's (nonparametric) tests for both diversity metrics (Shannon's diversity index and ACE) indicated varied results depending on the reference database used (Tables 1B and 2B). Of note, the museum specimens were significantly different among all pairwise comparisons examined for both Shannon's diversity index and ACE (Benjamini-Hochberg-adjusted p<0.05; α = 0.05). Pairwise comparisons between fresh specimen methods varied. The shotgun metagenomic (Rep200)-16S rRNA gene (fresh) comparison was significantly different for both diversity metrics whereas the shotgun metagenomic (WoL)-16S rRNA gene (fresh) comparison was not significantly different (p>0.05; α = 0.05) for both diversity metrics. The between shotgun metagenomic comparisons of WoL and Rep200 (fresh) were not significantly different for ACE (p = 0.232, α = 0.05), but were significantly different for Shannon's diversity (p = 0.0216, α = 0.05). The respective alpha diversity boxplots qualitatively compare the shotgun metagenomic sequencing data classified with the Rep200 database, shotgun metagenomic sequencing data classified with the WoL database, and 16S rRNA gene sequencing results classified with Greengenes (Fig 1A–1D).

## Mantel tests and procrustes analysis

Mantel tests and Procrustes analysis of fresh and museum-preserved specimens indicated no statistically significant correlations (i.e. there are differences between methods) between 16S rRNA gene (Greengenes) and shotgun metagenomic sequencing (both Rep200 and WoL) methods. Specifically, Mantel test Spearman's correlations (ρ) ranged from 0.139–0.358 (permutations = 999 for all comparisons) and associated p-values for all comparisons ranged from 0.158–0.711 (16S-Shotgun (WoL) comparison in Fig 2A and 2B; 16S-Shotgun (Rep200) comparison in S4A and S4B Fig). These results indicated no significant differences in Jaccard matrix distance correlations between 16S rRNA gene and shotgun metagenomic sequencing among the tested pairwise comparisons.

Additionally, Procrustes analysis (16S-Shotgun (WoL) comparison in Fig 3A and 3B; 16S-Shotgun (Rep200) comparison in S5A and S5B Fig) indicated significant differences between 16S and shotgun metagenomic results for museum but not fresh specimens. For fresh specimens both the 16S rRNA gene-WoL and 16S rRNA gene-Rep200 comparisons were not significant (p>0.05; α = 0.05). For museum specimens both the 16S rRNA gene-WoL and 16S rRNA gene-Rep200 comparisons were significantly different (p = 0.019 (of true $M^2$), p = 0.007 (of true $M^2$), respectively; α = 0.05). However, the 16S rRNA gene-WoL and 16S rRNA gene-Rep200 comparisons also had high $M^2$ values that could be interpreted as diminishing the significance of these results (True $M^2$ = 0.520, True $M^2$ = 0.481, respectively), as low $M^2$ values are generally needed to corroborate significant p-values. For the purposes of this study, they are interpreted as significant in the context of other statistical results presented.

## Taxonomic profiling

Taxa abundance bar plots indicated qualitative differences in taxa prediction between sequencing method (16S/Greengenes, shotgun/WoL, and shotgun/rep200) and specimen type (museum and fresh specimens of *R. pipiens)*. These qualitative differences indicated a general trend towards predictive divergence with increasing taxonomic resolution.

When examining Phyla-level resolution, all sequencing methods (16S/Greengenes, shotgun/WoL, and shotgun/rep200) had substantial overlap in their taxonomic predictions, with the Phyla Proteobacteria (Pseudomonodata), Actinobacteria, and Firmicutes being overrepresented as compared to other Phyla (S6 Fig). This trend was generally exhibited for both fresh specimens and museum specimens.

Changing the taxonomic classification to Genus-level resolution indicated wider discrepancies between sequencing methods and the fresh versus museum specimen comparison, broadly

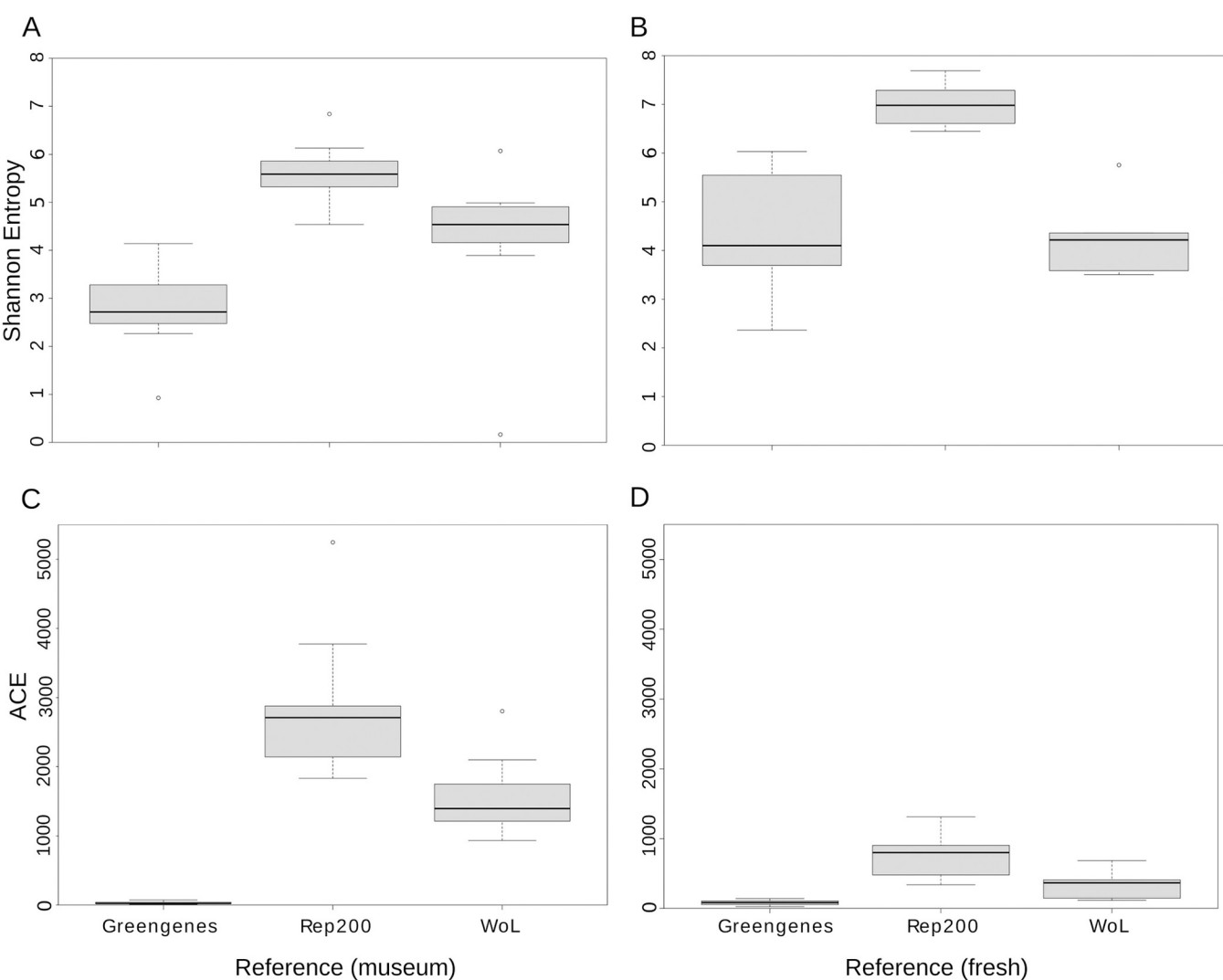

**Fig 1.** Boxplots of samples (n = 18) of the Shannon entropies for the different reference procedures from **(A)** museum specimens (n = 13) and **(B)** fresh specimens (n = 5) and the ACE values for the different reference procedures from **(C)** museum specimens (n = 13) and **(D)** fresh specimens (n = 5). The top and bottom of the boxes represent the upper and lower quartile boundaries, respectively. The black band is the median percentile. Bounds of the upper whisker is the default setting, calculated as min(max(x), Q_3 + 1.5 * IQR) and the bounds of the lower whisker is max(min(x), Q_1–1.5 * IQR).

visualized in the method-specific heatmaps (Figs 4 and S7). More specifically, 16S rRNA gene sequencing showed three Genera dominating the total community abundance in the museum specimens, whereas fresh specimens had more unique moderately represented Genera (16S and Shotgun (WoL) given in Fig 5A and 5B). This trend also held with shotgun metagenomic sequencing when comparing museum and fresh specimens with the Rep200 database (S6 Fig). The WoL database also showed few dominating Genera for both the fresh and museum specimens (Fig 5B), although these were classified differently, with *Acinetobacter* highly represented among the fresh specimens and *Pseudomonas* and *Enterobacter* highly represented among the museum specimen communities. *Pseudomonas*, *Enterobacter*, *Acinetobacter*, and *Pantoea* were however predicted as top ten Genera among all of the sequencing/analysis methods (Figs 4, 5, S6 and S7). Additionally, both shotgun metagenomic (WoL) and 16S rRNA gene sequencing methods exhibited concurrent sensitivity to a high prevalence of *Rickettsiella* in specimen 22967, which is an outlier from the other museum specimens.

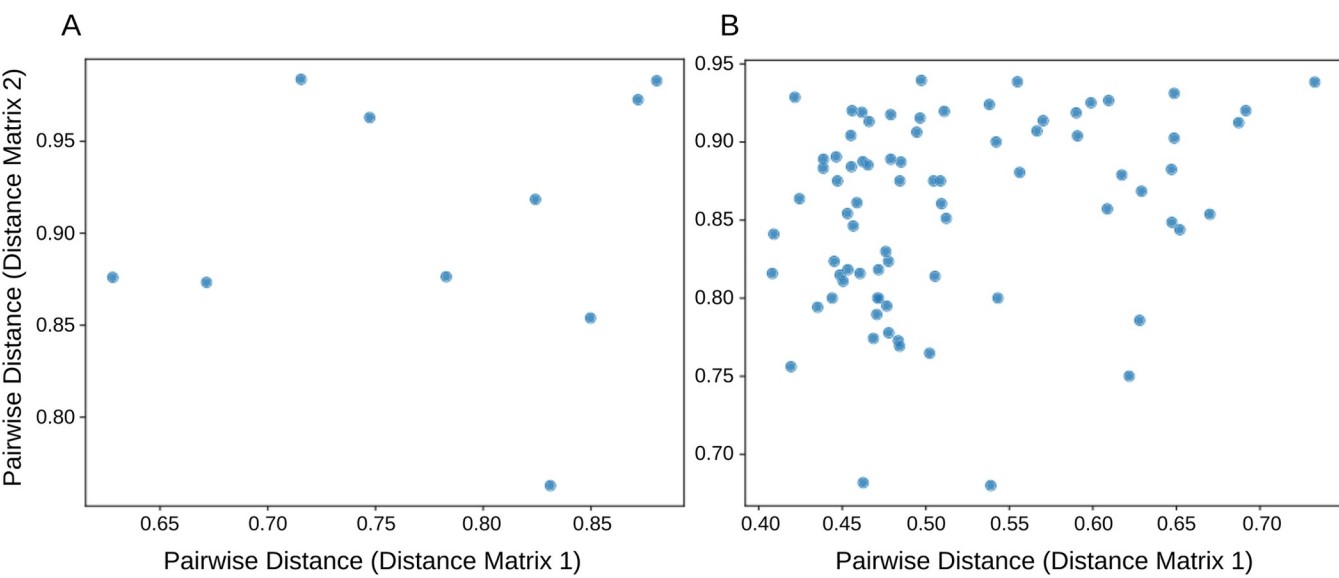

**Fig 2. Mantel tests of Jaccard distance matrices from shotgun metagenomic sequencing data (WoL databases) and 16S rRNA gene sequencing data (Greengenes database).** Fresh specimens are compared in Panel A and museum specimens are compared in panel B. **(A)** Spearman's rho = 0.139, p-value = 0.711. **(B)** Spearman's rho = 0.274, p-value = 0.243.

In addition to bacterial predictions, all databases used had Archaea present which was well represented among samples examined by both 16S rRNA gene and shotgun metagenomic sequencing, and also among both fresh and museum specimens. The presence of Euryarchaeota was confirmed from both the WoL and rep200 comparisons of the shotgun metagenomic sequencing data, which is in agreement with previously described amphibian samples

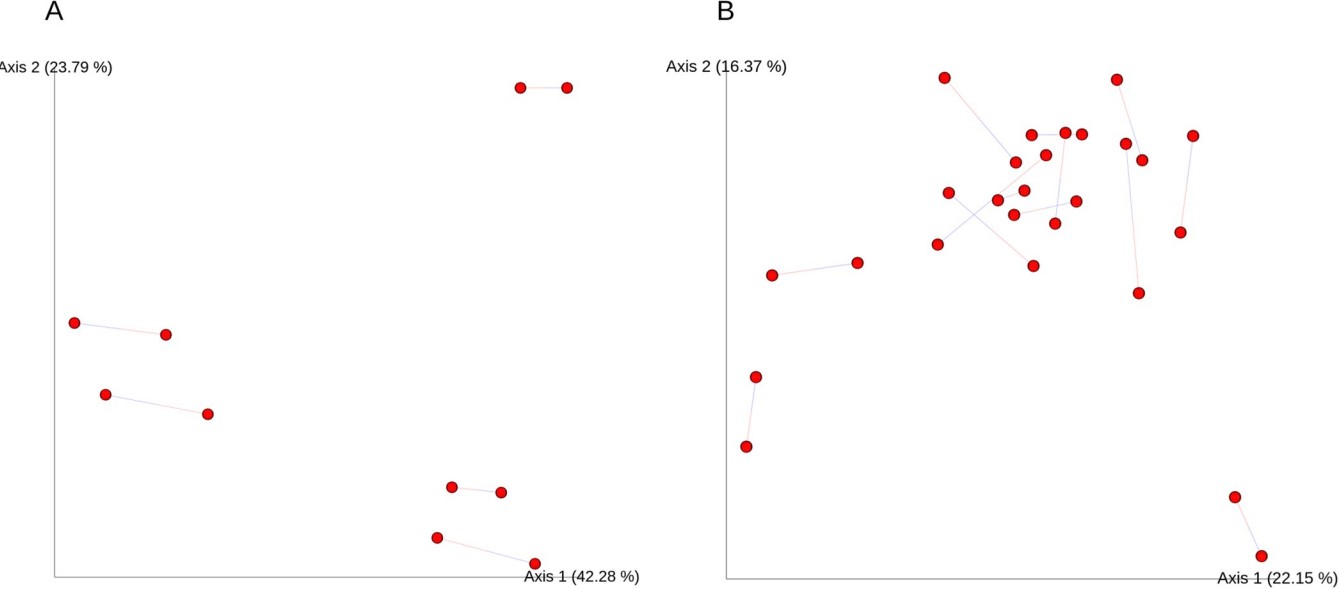

**Fig 3. Procrustes analysis with Jaccard distance-matrix derived principal coordinates.** The comparisons given are between: **(A)** Fresh specimen shotgun metagenomic sequencing (WoL database) and 16S rRNA gene sequencing (Greengenes database) with a true $M^2$ = 0.0498; p-value (of true $M^2$) = 0.475. **(B)** Museum-derived specimen shotgun metagenomic sequencing (WoL database) and 16S rRNA gene sequencing (Greengenes database). True $M^2$ = 0.520; p-value (of true $M^2$) = 0.019. A total of n = 999 Monte Carlo simulations was used for all comparisons.

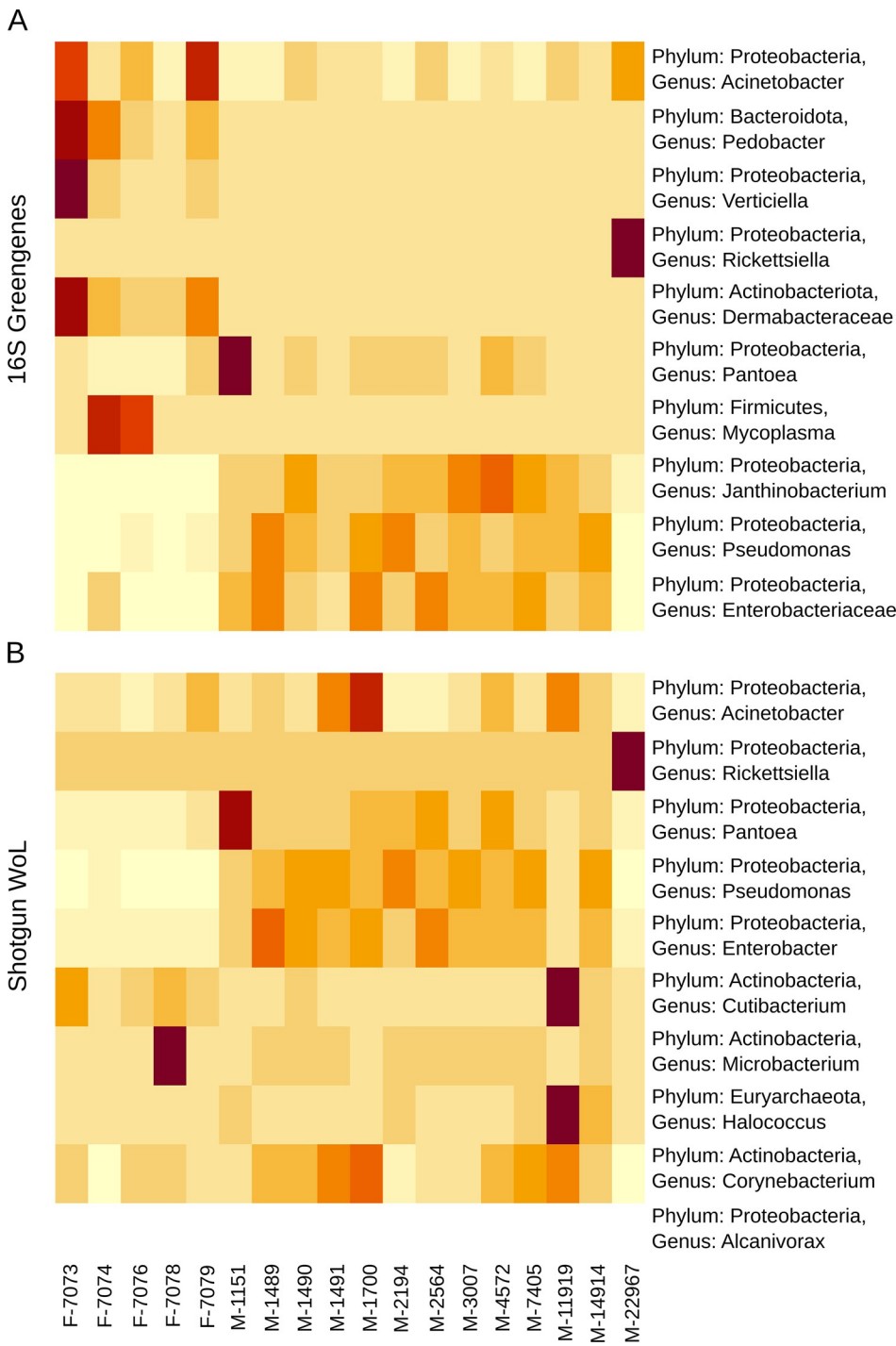

**Fig 4. Heatmap of the ten (n = 10) most highly represented Genera (of combined fresh and museum specimens) using (A) 16S rRNA gene and (B) Shotgun metagenomic (WoL).** Each row totals 100% (row normalized) with dark red approaching 100% and light yellow approaching 0%. For squares with <0.25%, 0 is assumed for purposes of visualization and normalization. For Genus-level groupings not having a single known ID, the next highest known taxonomic level is given. Specimen ID is given for each column in addition to specimen type: F (fresh) or M (museum).

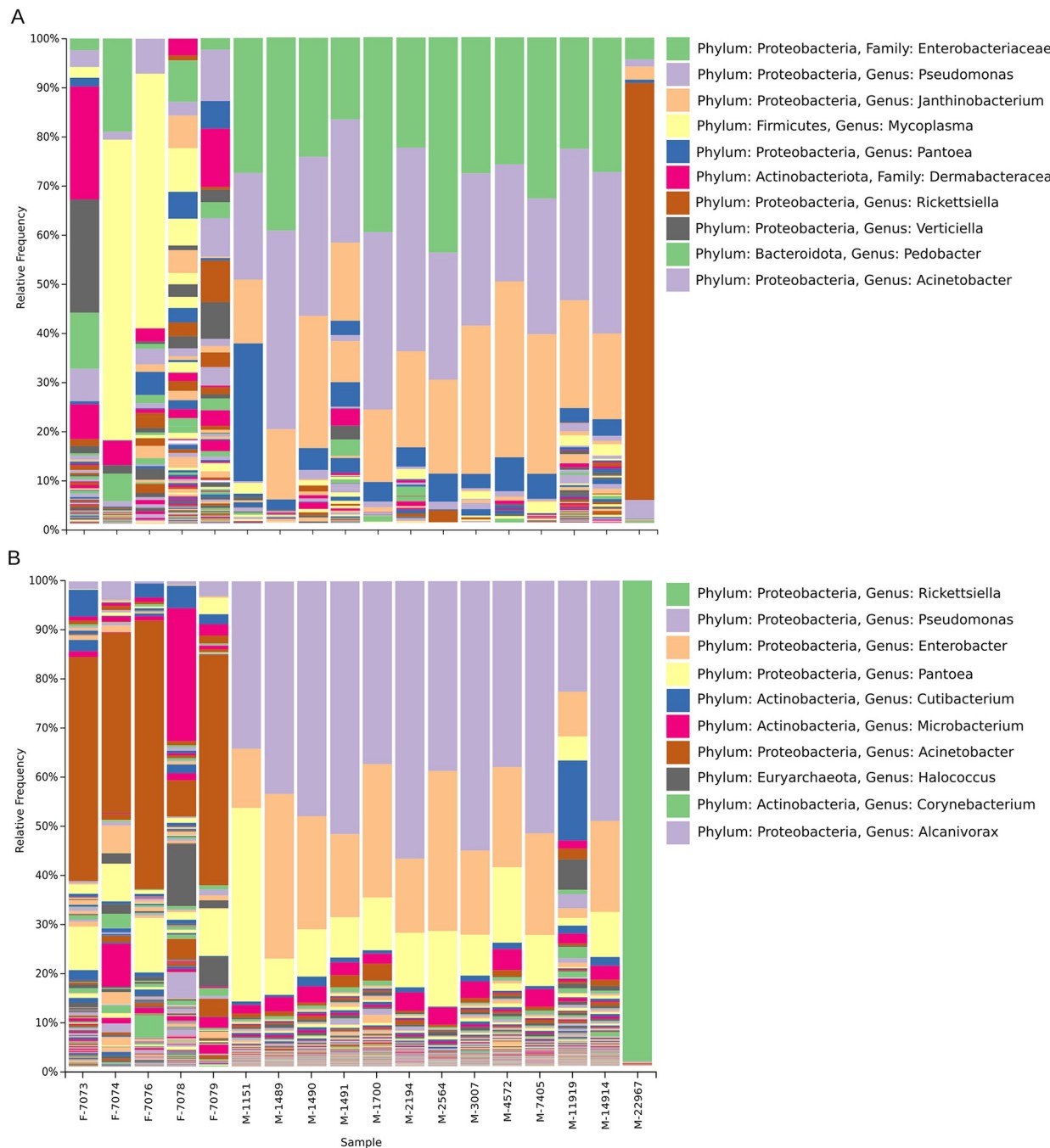

**Fig 5.** (A) Abundance barplot of all specimens (n = 18) with 16S rRNA gene sequencing derived taxonomies based on the Greengenes database. The top ten most commonly represented Genera are given in the legend. For Genus-level groupings not having a single known ID, the next highest known taxonomic level is given. Specimen ID is given for each column in addition to specimen type: F (fresh) or M (museum). (B) Abundance barplot of all specimens (n = 18) WoL database. The top ten most commonly represented Genera are given in the legend. For Genus-level groupings not having a single known ID, the next highest known taxonomic level is given. Specimen ID is given for each column in addition to specimen type: F (fresh) or M (museum).

[35] and possibly ubiquitous among amphibians. Other Phylum-level Archaeal identification included Crenarchaeota, of which there was agreement on presence in the top ten most abundant Phyla between 16S rRNA gene (Greengenes) and shotgun metagenomic (WoL), with

presence also confirmed in one sample by shotgun sequencing with Rep200 (although not in the top ten most represented). Genus-level taxonomic identification varied more considerably among sequencing methods and databases. Results other than Bacteria and Archaea were also produced by the shotgun metagenomics data categorized with Rep200 database. These results included limited classification of viruses, Protists, and Fungi (see Data Accessibility).

## Discussion

In this study, we examined differences in 16S rRNA gene and shotgun metagenomic sequencing derived nucleic acid datasets from freshly sampled and preserved museum specimens, hypothesizing that there would be differences between sequencing methods and between specimen type (museum or fresh). We confirmed our hypotheses by finding differences in diversity metrics and taxonomic predictions between 16S rRNA gene and shotgun metagenomic sequencing methods in both fresh and museum specimens, but with more pronounced differences in museum specimens. Alpha and beta diversity metric predictions and related statistical tests also exhibited some overlap between 16S rRNA gene and shotgun metagenomic sequencing methods in museum and fresh specimens among highly represented taxa.

Alpha diversity metrics differed significantly depending on the reference methods used, with museum specimens exhibiting significant differences between 16S rRNA gene and shotgun metagenomic sequencing (WoL and Rep200) for both diversity metrics (Shannon and ACE). Significant differences among fresh specimens were neither uniform nor as pronounced as museum specimens, and depended on the comparison being made and diversity metric chosen. The lack of significant alpha diversity differences seen in the shotgun metagenomic (WoL)-16S rRNA gene (fresh) comparison indicate that comparisons of gut microbiome datasets using differing sequencing methods are at times appropriate. This was corroborated in the case where specimen 22967 was flagged for high Rickettsial abundance, which may indicate a high infection rate at capture. However this was not seen for the Rep200 database indicating possible database-dependent sensitivity. Generally, this finding does not hold for the museum-derived microbiome samples. Mantel tests and Procrustes analysis of Jaccard distance matrices and derived PCoA results also indicate technical significance in the data generated by the compared methods. Specifically, Mantel tests showed no significant correlations among compared distances matrices, indicating differences between 16S rRNA gene and shotgun metagenomic methods for both fresh and museum-derived samples. Procrustes analysis also indicates significant differences among method for the museum specimens, but not for the fresh specimens.

In this study, the comparison of 16S rRNA gene and shotgun metagenomic derived data presented challenges for equitable comparison. A major consideration was in deciding which diversity metrics and reference databases to use for taxonomic comparison of the different datasets, specifically in referenced sequence abundance. To account for differences in abundance calculations of shotgun metagenomic and 16S rRNA gene sequencing, the Jaccard index of dissimilarity was used for efficacy of comparisons between sequencing strategies. This allowed taxonomic presence/absence determination between both strategies, as compared to abundance-linked diversity determinations (e.g. evenness in Shannon index). Abundance based diversity comparisons require correction or normalization procedures that are highly non-trivial when sourcing data or reference standards, and were outside of the scope of this study yet of interest for future work. The difficulty for correction or normalization arises due to copy numbers of sequences from 16S V4 sequences and those of taxonomically linked genomic regions represented in shotgun metagenomic sequencing data having uncorrelated or stochastic relationships. Where these abundance differences occur (and thus downstream diversity predictions), both methods would still be represented as either "present" or "absent"

for the referenced taxa using the Jaccard index of dissimilarity, regardless of abundance. The one caveat to this are those taxa that are only present in one method and not the other, which represents a quantifiable difference in the scheme utilized and presented in this study as results. For the purposes of this study, we therefore constrained our beta diversity comparisons to the Jaccard's index of dissimilarity as a diversity measure.

Similar considerations for taxa under-representation due to the degraded nature of museum-sourced nucleic acid datasets were made for the chosen alpha diversity metrics. Specifically, the ACE index was used which is designed to account for low abundance or under-representation applications. This was in addition to the commonly used Shannon's diversity index, which showed differences from ACE in a subset of comparisons and was thus indicative of the importance of alpha diversity metric choice when using degraded nucleic acids (such as those from fluid-preserved museum specimens).

The results of this study represent the first comparison of museum-derived microbiome data with two commonly used sequencing strategies for such analysis: targeted 16S rRNA gene sequencing and medium-depth shotgun metagenomic sequencing. Multiple reference database and analysis procedures were also compared and represent those commonly used in microbiome studies. As this was only a preliminary analysis of differences between sequencing types in museum specimen microbiomes, future work should aim to include samples from different Chordate groups (e.g. fish) which may vary in their preservation methods and subsequently have different results than those for amphibians. Future studies should also aim to understand degradation effects on different parts of the genome. Areas targeted by 16S rRNA gene sequencing may be more or less susceptible to formalin fixation or alcohol shearing as compared to shotgun sequencing. If such an effect were found and quantified, it may vary by species, but nonetheless be susceptible to corrective procedures in quality control and processing for improved comparisons between sequencing strategies. As sequencing technologies change and allow greater depth and lower costs, comparisons such as this should continue to ensure inter-method comparisons are justified and that the best methods are used. For those research questions that are focused on microbial community ecology and evolution spanning host-associated microorganisms outside of Bacteria and Archaea (i.e. 16S), shotgun metagenomic sequencing might be the preferred method due to the broad nucleic acid capture, and recognition of taxa other than Bacteria and Archaea (as seen with the Rep200 database used in this study generating limited results for viruses, fungi, and protozoa). Differences will also be dependent on the database used for taxonomic reconstruction or any downstream functional inference, such as the difference in represented taxa in the Rep200 (Archaea, Bacteria, Fungi, Protozoa, and viruses), WoL (Bacteria and Archaea), and Greengenes (Bacteria and Archaea) databases compared in this study. However, it is also important to make clear that the different Domains of taxa represented does not necessarily mean significant differences in downstream diversity comparisons, as was the case for the Rep200 –WoL fresh specimen comparison for ACE diversity.

Taken together, these results indicate that sequencing method choice is an important consideration when examining museum specimens—even more so than fresh specimens. These differences are in general agreement with previous work examining 16S rRNA gene and shotgun metagenomic sequencing methods showing differences between these methods and the commonly used downstream analysis pipelines [11,13,36]. The work presented here is therefore important in including museum specimens as a sample type that follows this broader trend.

Differences in results between these two sequencing methods will also be important in future efforts to systematically integrate molecular datasets with associated specimens in museum repositories [37]. How interpretation of any such integration will occur should be

thought about carefully due to implications including policy determination [38] and the trajectory of future studies that will build upon results of initial museum-associated microbiome studies.

## Supporting information

**S1 Fig. Examination of Kraken2 confidence threshold on reads classified for a subset of samples.** These data indicate a qualitative leveling-off at 0.8, which was subsequently used as a conservative confidence threshold (high confidence) so as to avoid incorrect classification of real microbial sequences as background data.
(TIF)

**S2 Fig. Panels show the quality profile of reads corresponding to each sample analyzed from 16S rRNA gene sequencing.** Quality score is given on the Y-axis and cycle number corresponding to read length is given on the X-axis. Reads were trimmed to 100bp based on these profiles, due to the quality decline for >100bp.
(TIF)

**S3 Fig. Rarefaction curve (generated in Qiita) with box plots representing the distribution of the Shannon alpha diversity metric for all samples combined (n = 18) at each even sampling depth.** The lower and upper whiskers of the box plot are the 9th and 91st percentiles of the distribution (respectively), while the lower and upper extents of the box are the 25th and 75th percentiles of the distribution (respectively). The horizontal bar through the middle of the box is the median of the distribution (i.e., the 50th percentile). Outlier points of these distributions are not shown. The line chart connects the median Shannon diversity value distribution across the sampling depths. If a sampling depth is higher than the number of sequences in a sample, that sample is not included in the rarefaction plot at that sampling depth.
(TIF)

**S4 Fig.** A–B. Mantel tests of Jaccard distance matrices from shotgun metagenomic sequencing data (Rep200 databases) and 16S rRNA gene sequencing data (Greengenes database). Fresh specimens are compared in Panel A and museum specimens are compared in panel B. (A) Spearman's rho = 0.358, p-value = 0.451. (B) Spearman's rho = 0.314, P-value = 0.158.
(TIF)

**S5 Fig.** A–B. Procrustes analysis with Jaccard distance-matrix derived principal coordinates. The comparisons given are between: **(A)** Fresh specimen shotgun metagenomic sequencing (Rep200 database) and 16S rRNA gene sequencing (Greengenes database) with a true $M^2$ = 0.0482; p-value (of true $M^2$) = 0.219. **(B)** Museum-derived shotgun metagenomic sequencing (Rep200 database) and 16S rRNA gene sequencing (Greengenes database) with a true $M^2$ = 0.481; p-value (of true $M^2$) = 0.007.
(TIF)

**S6 Fig.** A–D. Abundance barplots. The top ten most commonly represented Phyla or Genera given in the legend. For taxa-level groupings not having a single known ID, the next highest known taxonomic level is given. Specimen ID is given for each column in addition to specimen type: F (fresh) or M (museum). **(A)** Abundance barplot of all specimen (n = 18) shotgun metagenomic sequencing derived taxonomies based on the Rep200 database. The top ten most commonly represented Genera are given in the legend. **(B)** Abundance barplot of all specimen (n = 18) shotgun metagenomic sequencing derived taxonomies based on the WoL database. The top ten most commonly represented Phyla are given in the legend. **(C)** Abundance barplot of all specimen (n = 18) shotgun metagenomic sequencing derived taxonomies based on the

Rep200 database. The top ten most commonly represented Phyla are given in the legend. **(D)** Abundance barplot of all specimen (n = 18) 16S rRNA gene sequencing derived taxonomies based on the Greengenes database. The top ten most commonly represented Phyla are given in the legend.
(TIF)

**S7 Fig.** A–D. Heatmap of the ten (n = 10) most highly represented. **(A)** Phlya (of combined fresh and museum specimens) using 16S rRNA gene (Greengenes) sequencing **(B)** Phlya (of combined fresh and museum specimens) using Shotgun metagenomic (Rep200) sequencing **(C)** Phlya (of combined fresh and museum specimens) using Shotgun metagenomic (WoL sequencing **(D)** Genera (of combined fresh and museum specimens) using Shotgun metagenomic (Rep200) sequencing. Each row totals 100% (row normalized) with dark red approaching 100% and light yellow approaching 0%. For squares with <0.25%, 0 is assumed for purposes of visualization and normalization. For Phyla or Genus-level groupings not having a single known ID, the next highest known taxonomic level is given. Specimen ID is given for each column in addition to specimen type: F (fresh) or M (museum).
(TIF)

**S1 Table. Shotgun metagenomic sequencing raw read (forward and reversed separated) counts and stats generated from 'seqfu count'.**
(CSV)

**S2 Table. Per sample read counts of the 16S rRNA gene sequencing data.** Both raw counts and DADA2 filtered counts are given.
(CSV)

**S1 File.**
(CSV)

## Acknowledgments

Specimens used in this study were sourced from the Milwaukee Public Museum Vertebrate Zoology Collection, the Illinois Natural History Survey Herpetology Collection, and the University of Illinois Museum of Natural History Amphibian and Reptile Collection. Fresh samples were received from collection and subsequent dissection completed by Drew R. Davis. Specimens used in preliminary work to test dissection and swabbing methods were provided by Travis LaDuc from the University of Texas at Austin Herpetology Collection. Additionally, JM is thankful for advice and insights received that improved this manuscript during his time as a visiting scholar in the Knight Lab at the University of California, San Diego.

## Author Contributions

**Conceptualization:** Joseph D. Madison, Douglas C. Woodhams.

**Data curation:** Joseph D. Madison.

**Formal analysis:** Joseph D. Madison, Brandon C. LaBumbard, Douglas C. Woodhams.

**Funding acquisition:** Joseph D. Madison, Douglas C. Woodhams.

**Investigation:** Joseph D. Madison, Brandon C. LaBumbard, Douglas C. Woodhams.

**Methodology:** Joseph D. Madison, Brandon C. LaBumbard, Douglas C. Woodhams.

**Project administration:** Joseph D. Madison.

**Resources:** Joseph D. Madison.

**Software:** Joseph D. Madison, Brandon C. LaBumbard.

**Supervision:** Joseph D. Madison.

**Validation:** Joseph D. Madison.

**Visualization:** Joseph D. Madison.

**Writing – original draft:** Joseph D. Madison, Brandon C. LaBumbard, Douglas C. Woodhams.

**Writing – review & editing:** Joseph D. Madison, Brandon C. LaBumbard, Douglas C. Woodhams.

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
