## [Decision Letter · Decision Letter 0]

2 Jun 2023

PONE-D-23-13742Shotgun Metagenomics Describes Gut Microbiomes Better than Targeted 16S rRNA Gene Sequencing for Field Specimens and Preserved Museum SpecimensPLOS ONE

Dear Dr. Madison,

Thank you for submitting your manuscript to PLOS ONE. After careful consideration, we feel that it has merit but does not fully meet PLOS ONE’s publication criteria as it currently stands. Therefore, we invite you to submit a revised version of the manuscript that addresses the points raised during the review process. Please submit your revised manuscript by Jul 17 2023 11:59PM. If you will need more time than this to complete your revisions, please reply to this message or contact the journal office at plosone@plos.org. Please include the following items when submitting your revised manuscript:A rebuttal letter that responds to each point raised by the academic editor and reviewer(s). You should upload this letter as a separate file labeled 'Response to Reviewers'.A marked-up copy of your manuscript that highlights changes made to the original version. You should upload this as a separate file labeled 'Revised Manuscript with Track Changes'.An unmarked version of your revised paper without tracked changes. You should upload this as a separate file labeled 'Manuscript'.

We look forward to receiving your revised manuscript.

Kind regards,

Ruslan Kalendar

Academic Editor

PLOS ONE

Journal Requirements:

3. Please expand the acronym “NSF” (as indicated in your financial disclosure) so that it states the name of your funders in full.

"JDM is supported by the NSF Postdoctoral Research Fellowships in Biology Program (Award 1907311) and DCW by NSF Career and Infrastructure Awards (Award IOS-845634 and BII 2120084). Any opinions, findings, and conclusions or recommendations expressed in this material are those of the authors and do not necessarily reflect the views of NSF. Specimens used in this study were sourced from the Milwaukee Public Museum Vertebrate Zoology Collection, the Illinois Natural History Survey Herpetology Collection, and the University of Illinois Museum of Natural History Amphibian and Reptile Collection. Fresh samples were received from collection and subsequent dissection completed by Drew R. Davis. Additionally, JM is thankful for advice and insights received that improved this manuscript during his time as a visiting scholar in the Knight Lab at the University of California, San Diego."

"JDM is supported by the NSF Postdoctoral Research Fellowships in Biology Program (Award 1907311; www.nsf.gov) and DCW by NSF Career and Infrastructure Awards (Award IOS-845634 and BII 2120084; www.nsf.gov). Any opinions, findings, and conclusions or recommendations expressed in this material are those of the authors and do not necessarily reflect the views of NSF. NSF had no role in study design, data collection and analysis, decision to publish, or preparation of the manuscript."

Reviewers' comments:

Reviewer's Responses to Questions

**Comments to the Author**

1. Is the manuscript technically sound, and do the data support the conclusions?

Reviewer #1: Partly

Reviewer #2: Partly

2. Has the statistical analysis been performed appropriately and rigorously? 

Reviewer #1: Yes

Reviewer #2: Yes

3. Have the authors made all data underlying the findings in their manuscript fully available?

Reviewer #1: Yes

Reviewer #2: Yes

4. Is the manuscript presented in an intelligible fashion and written in standard English?

Reviewer #1: No

Reviewer #2: Yes

5. Review Comments to the Author

Reviewer #1: 

### Summary

In this study, Madison et. al. present a comparative analysis of animal-associated gut microbiomes from a variety of sources using a variety of sequencing and software methods. Overall, this is a worthy effort - the topic is interesting, and the data collected can certainly make an important contribution to the field. However, the overall message of the manuscript is somewhat garbled, and I think substantial revision is necessary to make the claims clear enough to evaluate fully. To be clear - I do not expect the authors to perform any additional experiments, sample collections, or sequencing. Rather, the presentation of data (both narratively and in figures) needs a bit of reworking.

As I see it, the authors are jumping between a number of different comparisons:

1. Museum specimens vs samples from freshly caught specimens

2. Metagenomic vs Amplicon sequencing

3. Different metagenomic databases / profiling approaches.

However, the specific comparisons in any given figure / analysis are mixed together, and it is generally unclear which comparison is germaine for a given statistic or plot. Given that there is a robust literature comparing metagenomic and amplicon sequencing, as well as benchmarking different metagenomic profiling methods, the authors should lean into the museum samples, and make this clear. For example - it's known that mgx provides better taxonomic resolution than amplicon, would you expect this to be the case in museum samples that are substantially degraded? I would expect amplicon sequencing to be at an even greater disadvantage, especially since you're using kmer-based methods for the metagenomics (need much smaller overlap). Can you show this?

In most cases, I think that (3) above should be de-emphasized. The authors should pick which method makes the most sense, and stick with it. Alternatively, present **one** figure that directly compares the databases in museum vs fresh samples, and explains advantages / disadvantages of each. Comparing 2 sequencing methods, one of which has 2 databases, and treating these as 3 independent categories is both confusing and statistically unsound.

In figures, things being compared should be included in the same plot to make that comparison clear (indeed possible). For example, in figure 2, if the goal is to compare fresh to museum samples, Combine A/B or C/D, and color dots by the source. If the goal is to compare metagenomic databases, combine A/C and C/D. As another example, figures 4/5 could be combined, with profiles from the same sample plotted side-by-side. At minumum, the same colors should be used for the same taxon accross plots of that type.

Finally, the claim in the title should be reined in a bit. Given that you do are not comparing metagenomes of known composition, it's tough to say that one method "Describes Gut Microbiomes Better" than the other. I certainly assume that would be true, but nothing in this manuscript demonstrates that directly. You can make claims about observed diversity, or any number of other findings.

### Minor points / suggestions

1. It would be worthwhile to include some additional justifaction for the use of jaccard distance throughout. Most microbial community studies use distance / dissimilarity metrics that take relative abundance of taxa into account (eg Bray Curtis / UniFrac). I don't think it's wrong to use jaccard persay, but especially when comparing amplicon to mgx, since the composition of references in the database are substantially different, if there are a bunch of low-abundance taxa that are present in one but not the other, it will have an outsized impact on Jaccard distance

2. I already mentioned a suggestion for the abundance bar plots, but the sortin on them is also a bit weird. You can/should use hierarchical clustering to determine the order (if you **are** using it, look into the "optimal leaf ordering" algorithm). It's also quite hard visually to compre "F" vs "M" in the text - maybe make the museum / fresh sample labels different colors?

3. The PDF I downloaded has a number of strange display quirks, eg different fonts (the primary one of which is quite low resolution), and many of the plots are covered in dark gray squares (looks like a PDF embedding issue, since some of the alignments are off as well). I don't know if this is the result of PDF processing on the website or what was uploaded, but it was a bit of a challenge to read.

4. Some kind of summary of quality scores (Figure S2) would be helpful, rather than (or in addition to) showing each sample separately. In particular, it would be useful to see a comparison of these data between amplicons from fresh vs museum samples.

5. You mentioned that you have museum samples from over 100 year span, which is awsome - are there any trends you can see over time?

Reviewer #2: 

The authors present a comparison of two different sequencing methods (metagenomic vs metabarcoding) with three different databases (two for metagenomic, one for metabarcoding) and two different types of samples (fresh vs. museum) using frogs as study species. Overall, I think the topic is relevant and of wider interest. However, the technical novelty is relatively limited since no new protocols are being proposed. Nevertheless, it is a worthwhile general discussion to be had.

Some of my concerns are the low sequencing depth of the 16S data, which is further exacerbated by rarefying the data to 3,000 reads. I furthermore disagree with some of the statements made (see below), but it does provide an interesting discussion point. However, some of the statements made in the discussion about the comparability of the two sequencing methods should maybe be revised. Lastly, the figures are of very low quality and should be improved.

Detailed comments

Line 46: I would say the relationship is reversed. This makes it sound like nucleic acid sequencing in itself is the aim, and museum specimens facilitate that. But what should come first is the question. And to address the question, the sequencing of museum specimens is used.

Lines 57-59: These are very broad statements; it would be nice to have some examples on how exactly this would be of use.

Lines 65-66: high-throughput sequencing has been around since 2006. I think we are way past the “advent of HTS”. I think it’s time to accept HTS as an established method and not keep referring to it as something novel.

Lines 71-73: it might be worth it to spend a couple lines outlining the differences found is fresh specimens here.

Line 90: I’m not sure if I understand this sentence. How does using one frog per decade allow to capture temporal degradation and storage effects? I guess you will get a range of results, but it wont be possible to attribute any changes to those two factors.

Line 147: Which flow cell type was used?

Line 148: there is a digit missing in this number “23,844,08”. And it’s better to use comma separators to mark thousands for these numbers: “range: 9108413–54875278”

Lines 160-162: It’s not quite clear what the authors are saying here. What kind of matches were removed and for what purpose? (It becomes clearer later, but it should be clarified upfront)

Line 170: it might have been better to first remove adapters before doing host removal. Although the effect is possibly minor.

Line 175: What was analysed? You give more details below for 16S, but this is missing here. Maybe move this information to the combined section for both analyses so readers can compare more easily? This comparison is very important in assessing the outcomes of this publication.

Line 181: it’s unnecessary to say “using the same protocol”. Just mention that the metabarcoding was done on the exact same extracted DNA samples.

Line 185: “sample amplicons”: I assume you are talking about the two replicate/duplicate PCR reactions done for each sample, and not all amplicons?

Lines 191-192: if a 300 cycle cartridge was used, the reads are already 150 bp in length. How could the be trimmed to 150 bp?

Lines 194-195: I’m not sure if a complete removal of ASV found in the negative samples is a good idea. One of the most likely sources of contamination in this process are the samples being sequenced. If now ASV are completely removed that way, the final diversity will be heavily biased. How rare or how abundant were the ASV that were removed? There are contamination removal tools that do a better job than just removing ASV completely.

Lines 196: 3,000 reads sounds very low to me nowadays. I often find that samples at such a low read depth show reduced diversity.

Lines 212-213: to analyse beta diversity between the two sequencing methods, the count tables need to be merged. How was this done? If not, then only beta diversity within each method can be assessed.

Lines 221-222: I have a feeling another large source of variation are the two different approaches for taxonomic assignment / databases used. Even within 16S sequencing, differences are expected using GG or SILVA, and this will be the case here to. I think it will be worthwhile to at least briefly address this point.

Lines 229-230 (and 237-238): Was any grouping by taxonomy done? If not, then the taxonomic assignment method and reference database for metabarcoding won’t have any effect on the result.

Figure 1: I realise that these wont be the final figures; however, the quality is quite bad to the effect that the labels can’t be read properly.

Figure 2: I’m not sure if the figure/analysis as presented is useful for the interpretation the authors want to make. It would rather see a classical nMDS or PCoA.

Figure 3: Stationary 3D figures are a really bad idea. They are incredibly difficult to interpret properly without being able to manipulate the angle of the 3D space.

Line 273-275: Unsurprisingly. One would expect a larger difference at the lower taxonomic levels.

Line 275: “comp16ared” should be “compared”

Lines 283-284: You are referring to Figure 6 before referring to Figure 5. This should be sequential.

Lines 287-289: More importantly than being an outlier, shotgun (WoL) and metabarcoding (GG) agreeing on the rickettsia abundance while shotgun (Rep200) disagrees is an interesting finding. It makes me question the validity of the Rep200 analysis, since this is a very obvious result that should have been recovered too. I hope to find a discussion of this later in the Discussion part. Although, rickettsiales could also potentially mean another case of mitochondrial contamination.

Figures 4, 5, 6: It would be easier if these three figures were maybe combined into one multipanel figure, and also the order of the X axis being made the same in both. As it’s standing, it is very difficult to compare the three methods. (also, the dimensions of the figures differ)

Heatmaps: not sure if they add anything to the argument. (To the point where the only mention of figure 7 is with figures 4-6 in a single sentence.)

Lines 317-319: A comparison is not as straight forward as the authors make it seem. Especially looking at the diversity measures, I would not be inclined to compare between shotgun (WoL and 16S metabarcoding results. The authors state “at times” but when can they really be sure the time is right? Yes, the bar graph shows strong similarity (as far as I can see), but I don’t think this is enough cause.

Lines 333-335: It think it’s still possible to compare. It’s not a bias free comparison, but that’s not the point. You are not trying to compare Treatment A done with metagenomics with Treatment B done with 16S, I agree that would be impossible due to the non-trivial problems. However, comparing how metagenomics and 16S differ in comparing Treatment A and B is a valid analysis, that is independent of these abundance-linked issues. These issues are part of the nature of either sequencing method and are integral to thecomparison.

Supplementary Table 2: since you used DADA2, it would be interesting to see the full count table DADA2 produces for read filtering quality control, not just raw counts.

6. PLOS authors have the option to publish the peer review history of their article (what does this mean?). If published, this will include your full peer review and any attached files.

Reviewer #1: **Yes: **Kevin S. Bonham

Reviewer #2: No

---

## [Author Response · Author response to Decision Letter 0]

1 Aug 2023

Also attached in cover letter.

Journal Requirements:

https://journals.plos.org/plosone/s/file?id=wjVg/ and 

https://journals.plos.org/plosone/s/file?id=ba62/

We have ensured the revised manuscript meets style requirements.

We have moved grant information from Acknowledgements to Funding Information.

3. Please expand the acronym “NSF” (as indicated in your financial disclosure) so that it states the name of your funders in full.

We have spelled out U.S. National Science Foundation in the acknowledgments section.

All data referenced in the manuscript is now available at their corresponding repositories, and this is reflected in the Data Availability Statement.

 "JDM is supported by the NSF Postdoctoral Research Fellowships in Biology Program (Award 1907311) and DCW by NSF Career and Infrastructure Awards (Award IOS-845634 and BII 2120084). Any opinions, findings, and conclusions or recommendations expressed in this material are those of the authors and do not necessarily reflect the views of NSF. Specimens used in this study were sourced from the Milwaukee Public Museum Vertebrate Zoology Collection, the Illinois Natural History Survey Herpetology Collection, and the University of Illinois Museum of Natural History Amphibian and Reptile Collection. Fresh samples were received from collection and subsequent dissection completed by Drew R. Davis. Additionally, JM is thankful for advice and insights received that improved this manuscript during his time as a visiting scholar in the Knight Lab at the University of California, San Diego."

 "JDM is supported by the NSF Postdoctoral Research Fellowships in Biology Program (Award 1907311; www.nsf.gov) and DCW by NSF Career and Infrastructure Awards (Award IOS-845634 and BII 2120084; www.nsf.gov). Any opinions, findings, and conclusions or recommendations expressed in this material are those of the authors and do not necessarily reflect the views of NSF. NSF had no role in study design, data collection and analysis, decision to publish, or preparation of the manuscript."

We have moved grant information from Acknowledgements to Funding Information.

The Acknowledgments section should read:

Acknowledgements

Specimens used in this study were sourced from the Milwaukee Public Museum Vertebrate Zoology Collection, the Illinois Natural History Survey Herpetology Collection, and the University of Illinois Museum of Natural History Amphibian and Reptile Collection. Fresh samples were received from collection and subsequent dissection completed by Drew R. Davis. Additionally, JM is thankful for advice and insights received that improved this manuscript during his time as a visiting scholar in the Knight Lab at the University of California, San Diego. 

The Funding Statement should read:

Funding Statement

JDM is supported by the U.S. National Science Foundation Postdoctoral Research Fellowships in Biology Program (Award 1907311; www.nsf.gov) and DCW by U.S. National Science Foundation Career and Infrastructure Awards (Award IOS-845634 and BII 2120084; www.nsf.gov). Any opinions, findings, and conclusions or recommendations expressed in this material are those of the authors and do not necessarily reflect the views of the U.S. National Science Foundation. The U.S. National Science Foundation had no role in study design, data collection and analysis, decision to publish, or preparation of the manuscript.

We have moved the ethics statement to the Methods section.

7. Please include captions for your Supporting Information files at the end of your manuscript, and update any in-text citations to match accordingly. Please see our Supporting Information guidelines for more information: http://journals.plos.org/plosone/s/supporting-. 

Captions for Supporting Information files have been added at the end of the manuscript.

 Reviewers' comments:

 Reviewer #1: 

 ### Summary

In this study, Madison et. al. present a comparative analysis of animal-associated gut microbiomes from a variety of sources using a variety of sequencing and software methods. Overall, this is a worthy effort - the topic is interesting, and the data collected can certainly make an important contribution to the field. However, the overall message of the manuscript is somewhat garbled, and I think substantial revision is necessary to make the claims clear enough to evaluate fully. To be clear - I do not expect the authors to perform any additional experiments, sample collections, or sequencing. Rather, the presentation of data (both narratively and in figures) needs a bit of reworking.

We thank the reviewers for their valuable suggestions to revise the manuscript and respond to each below, in bold.

As I see it, the authors are jumping between a number of different comparisons:

 1. Museum specimens vs samples from freshly caught specimens

 2. Metagenomic vs Amplicon sequencing

 3. Different metagenomic databases / profiling approaches.

Correct, all three of these comparisons are possible and were made with our dataset.

 However, the specific comparisons in any given figure / analysis are mixed together, and it is generally unclear which comparison is germaine for a given statistic or plot. Given that there is a robust literature comparing metagenomic and amplicon sequencing, as well as benchmarking different metagenomic profiling methods, the authors should lean into the museum samples, and make this clear. 

We agree and thus have included “Preserved Museum Specimens” as part of the title. We have now emphasized this in the abstract, clarified the introduction section, and emphasized the museum specimens again in the discussion. As amphibian museum specimens are the focus of this study, the freshly sampled wild samples are a comparison group. We have also reformatted figures to make the comparisons being presented more clear, with emphasis on the the museum-fresh specimen comparison and the shotgun (WoL) – 16S comaprison. Figures comparing different shotgun databases have been put into supplemental.

For example - it's known that mgx provides better taxonomic resolution than amplicon, would you expect this to be the case in museum samples that are substantially degraded? I would expect amplicon sequencing to be at an even greater disadvantage, especially since you're using kmer-based methods for the metagenomics (need much smaller overlap). Can you show this?

Shotgun generally provides more complex alpha diversity representation (and we showed that here for museum specimens, which to this point was an unknown). To probe the question posed by the reviewer, the study would have to look at degradation patterns/rates in different areas of the genome. One could then gauge if regions targeted by 16S sequencing are more susceptible to degradation effects as compared to shotgun sequencing. While an interesting future direction, this was not the goal of this study so is not presented here. We have included additional language to suggest this as a future study in the discussion.

In most cases, I think that (3) above should be de-emphasized. The authors should pick which method makes the most sense, and stick with it. Alternatively, present **one** figure that directly compares the databases in museum vs fresh samples, and explains advantages / disadvantages of each. Comparing 2 sequencing methods, one of which has 2 databases, and treating these as 3 independent categories is both confusing and statistically unsound.

We agree that (3) above can be de-emphasized and have moved most of the database comparison figures to supplemental, to deemphasize, and will keep the primary comparison with 16S and WoL for museum and fresh specimens (as WoL has taxonomic representations more similar to the Greengenes for the 16S data; i.e. no viruses, fungi, etc., which we make note of in the manuscript). We have also edited the results and discussion section to reflect this change. However, we do not consider the comparisons made to be statistically unsound.

In figures, things being compared should be included in the same plot to make that comparison clear (indeed possible). For example, in figure 2, if the goal is to compare fresh to museum samples, Combine A/B or C/D, and color dots by the source. If the goal is to compare metagenomic databases, combine A/C and C/D. As another example, figures 4/5 could be combined, with profiles from the same sample plotted side-by-side. At minumum, the same colors should be used for the same taxon accross plots of that type.

Our questions of focus are (1) and (2), as noted above. Thus, we have combined figures comparing museum vs. fresh samples in one figure for amplicon-based data, and one figure for metagenomic data. These figures are all based on the WoL and Greegenes database. Colors vary by graph as matching taxa are not represented in each figure, and exact matching is not possible. We have provided additional interactive files for readers interested in changing figure parameters, different colors, etc.

 Finally, the claim in the title should be reined in a bit. Given that you are not comparing metagenomes of known composition, it's tough to say that one method "Describes Gut Microbiomes Better" than the other. I certainly assume that would be true, but nothing in this manuscript demonstrates that directly. You can make claims about observed diversity, or any number of other findings.

We agree and have changed the wording of the title to clarify and describe which methods provide the greatest diversity/taxonomic resolution.

 ### Minor points / suggestions

 1. It would be worthwhile to include some additional justification for the use of jaccard distance throughout. Most microbial community studies use distance / dissimilarity metrics that take relative abundance of taxa into account (eg Bray Curtis / UniFrac). I don't think it's wrong to use jaccard persay, but especially when comparing amplicon to mgx, since the composition of references in the database are substantially different, if there are a bunch of low-abundance taxa that are present in one but not the other, it will have an outsized impact on Jaccard distance.

While we agree that Jaccard is not the most widely used metric for amplicon based studies, we use it here because it allows for comparison between metagenomic and amplicon results in an abundance-independent manner. As noted in the methods section, the low-abundance taxa are filtered from the 16s datasets by rarefaction. Rarefaction was not used for shotgun per say, but different forms of clean-up/quality control were utilized. This difference is inherent to the data types being compared and discussed in the manuscript (e.g., rarefaction of shotgun data is not a well developed technique, as compared to 16S). Including abundance based metrics would be considerably more fraught and subject to highly non-trivial error propagation correction techniques that are not well developed.

 2. I already mentioned a suggestion for the abundance bar plots, but the sorting on them is also a bit weird. You can/should use hierarchical clustering to determine the order (if you **are** using it, look into the "optimal leaf ordering" algorithm). It's also quite hard visually to compre "F" vs "M" in the text - maybe make the museum / fresh sample labels different colors?

We agree, and have now re-ordered the taxa numerically for consistency, and also added larger x-axis labels indicating museum (M) vs. Fresh (F) samples below the sample names. This should help readers better view and compare between figures.

 3. The PDF I downloaded has a number of strange display quirks, eg different fonts (the primary one of which is quite low resolution), and many of the plots are covered in dark gray squares (looks like a PDF embedding issue, since some of the alignments are off as well). I don't know if this is the result of PDF processing on the website or what was uploaded, but it was a bit of a challenge to read.

We will be sure to double check the final figure files and also upload figures separately so these errors don’t translate to publication.

 4. Some kind of summary of quality scores (Figure S2) would be helpful, rather than (or in addition to) showing each sample separately. In particular, it would be useful to see a comparison of these data between amplicons from fresh vs museum samples.

We have added an additional column to this supplemental table, as suggested by the previous reviewer. Summary statistics are also given (both 16S and shotgun).

 5. You mentioned that you have museum samples from over 100 year span, which is awsome - are there any trends you can see over time?

This is not the focus of this manuscript, but to answer your question, a larger sample set based on the best methods described here, will describe the time trends found in the data. We are glad you find the study interesting!

 Reviewer #2: 

 The authors present a comparison of two different sequencing methods (metagenomic vs metabarcoding) with three different databases (two for metagenomic, one for metabarcoding) and two different types of samples (fresh vs. museum) using frogs as study species. Overall, I think the topic is relevant and of wider interest. However, the technical novelty is relatively limited since no new protocols are being proposed. Nevertheless, it is a worthwhile general discussion to be had.

We thank the reviewer for their interest in this study, and agree that the novelty is in using amphibian museum samples to determine the best methods to describe microbial communities, rather than in development of novel analytical protocols.

 Some of my concerns are the low sequencing depth of the 16S data, which is further exacerbated by rarefying the data to 3,000 reads. 

3,000 16S rRNA reads is quite good for amphibian bacterial communities. Other studies rely on many fewer reads (e.g. Mulla, L., & Hernández‐Gómez, O. (2023). Wildfires disturb the natural skin microbiota of terrestrial salamanders. Environmental Microbiology. AND García-Sánchez, J. C., Arredondo-Centeno, J., Segovia-Ramirez, M. G., Tenorio Olvera, A. M., Parra-Olea, G., Vredenburg, V. T., & Rovito, S. M. (2022). Factors influencing bacterial and fungal skin communities of montane salamanders of Central Mexico. Microbial Ecology, 1-17.), and rarefaction plots for the 16S data indicate that most of the diversity is accounted for. We have also included a rarefaction plot of Shannon diversity as a Supplemental figure in the revised manuscript).

I furthermore disagree with some of the statements made (see below), but it does provide an interesting discussion point. However, some of the statements made in the discussion about the comparability of the two sequencing methods should maybe be revised. 

We have revised the discussion for clarity and note that the purpose of this study is 1-3 described above, but with a focus on 1-2. . We have better highlighted the differences in taxonomic resolution and overall diversity by putting less emphasis on the shotgun database comparison (although still important) and comparing museum and fresh samples by both amplicon and metagenomic sequencing. Our conclusion is that shotgun sequencing provides the best taxonomic resolution with the greatest diversity for museum specimens.

Lastly, the figures are of very low quality and should be improved.

We have revised the figures as suggested by Reviewer 1.

 Detailed comments

 Line 46: I would say the relationship is reversed. This makes it sound like nucleic acid sequencing in itself is the aim, and museum specimens facilitate that. But what should come first is the question. And to address the question, the sequencing of museum specimens is used.

Revised, as suggested.

 Lines 57-59: These are very broad statements; it would be nice to have some examples on how exactly this would be of use.

Two references to support this claim/statement have been added.

 Lines 65-66: high-throughput sequencing has been around since 2006. I think we are way past the “advent of HTS”. I think it’s time to accept HTS as an established method and not keep referring to it as something novel.

Agreed. The intent of this statement was not to suggest high-throughput sequencing is a new technology. We have therefore eliminated the relevant language to clarify our meaning.

 Lines 71-73: it might be worth it to spend a couple lines outlining the differences found is fresh specimens here.

Additional language has been added to convey the general finding that resolution is higher with shotgun sequencing.

 Line 90: I’m not sure if I understand this sentence. How does using one frog per decade allow to capture temporal degradation and storage effects? I guess you will get a range of results, but it wont be possible to attribute any changes to those two factors.

We have revised this statement for clarity to note degradation effects due to storage time; we are not talking about two factors. We also note that this is not a main focus of this study and a larger sampling effort will deal with these temporal trends in museum samples. Our purpose here was to account for the variation that *may* have been found with museum specimens of differing ages. We have also changed the language accordingly, so as to make it clear we are not comparing two variables at once.

 Line 147: Which flow cell type was used?

S4 flow cell was used. This information has been added.

 Line 148: there is a digit missing in this number “23,844,08”. And it’s better to use comma separators to mark thousands for these numbers: “range: 9108413–54875278”

23,844,08 has been changed to 23,844,087. Commas have also been added.

 Lines 160-162: It’s not quite clear what the authors are saying here. What kind of matches were removed and for what purpose? (It becomes clearer later, but it should be clarified upfront)

We have added additional language here to make the goal more clear, as suggested. Specifically, that this step was to control for background contamination.

 Line 170: it might have been better to first remove adapters before doing host removal. Although the effect is possibly minor.

We believe this wouldn’t have an effect and have kept as is.

 Line 175: What was analysed? You give more details below for 16S, but this is missing here. Maybe move this information to the combined section for both analyses so readers can compare more easily? This comparison is very important in assessing the outcomes of this publication.

Have altered text to clarify that species-level feature-table files (.qza format) were used from Woltka output. This is also shown in the publicly available Qiita study.

 Line 181: it’s unnecessary to say “using the same protocol”. Just mention that the metabarcoding was done on the exact same extracted DNA samples.

Revised, as suggested.

 Line 185: “sample amplicons”: I assume you are talking about the two replicate/duplicate PCR reactions done for each sample, and not all amplicons?

Revised for clarity.

 Lines 191-192: if a 300 cycle cartridge was used, the reads are already 150 bp in length. How could the be trimmed to 150 bp?

This was single-end sequencing, and this has been clarified here. Supplemental, quality figures indicate the read length, and drop-off in quality necessitating trimming. 150 bp was chosen due to drop-off in quality at this point, and for enhanced comparability with the shotgun data.

 Lines 194-195: I’m not sure if a complete removal of ASV found in the negative samples is a good idea. One of the most likely sources of contamination in this process are the samples being sequenced. If now ASV are completely removed that way, the final diversity will be heavily biased. How rare or how abundant were the ASV that were removed? There are contamination removal tools that do a better job than just removing ASV completely.

Abundances were the same, justifying the removal. The supplemental file including these comparisons has been added. Contamination removal tools are generally designed for non-amphibian hosts (e.g. humans, rats) necessitating manual inspection/removal.

 Lines 196: 3,000 reads sounds very low to me nowadays. I often find that samples at such a low read depth show reduced diversity.

See response above. 

 Lines 212-213: to analyse beta diversity between the two sequencing methods, the count tables need to be merged. How was this done? If not, then only beta diversity within each method can be assessed.

This process has been described in more detail. Essentially, the comparison is of distance matrices of the same type (Jaccard). The differences in distances are then compared as the subject of the analysis.

 Lines 221-222: I have a feeling another large source of variation are the two different approaches for taxonomic assignment / databases used. Even within 16S sequencing, differences are expected using GG or SILVA, and this will be the case here too. I think it will be worthwhile to at least briefly address this point.

We agree as this was one of our questions. Can address this more if necessary in the introduction. Will include figures in the supplemental, as previously mentioned for reviewer 1.

 Lines 229-230 (and 237-238): Was any grouping by taxonomy done? If not, then the taxonomic assignment method and reference database for metabarcoding won’t have any effect on the result.

We are not clear as to the question. Grouping of reads by taxonomy - as we understand the reviewer - was not done before statistical analysis. We believe that the assignment method and reference database chosen for taxonomic profiling of processed sequencing data would indeed affect observed alpha diversity. Previous studies discussing database effects are cited in the introduction.

 Figure 1: I realise that these wont be the final figures; however, the quality is quite bad to the effect that the labels can’t be read properly.

The figures have been revised for quality.

 Figure 2: I’m not sure if the figure/analysis as presented is useful for the interpretation the authors want to make. It would rather see a classical nMDS or PCoA.

We have revised the figure (some moved to supplemental). This should aid in interpretation. The Procrustes analysis aims to find the Frobenius norm, which is the analysis we wish to make in comparing distance matrices. Specifically, the matrices are Jaccard PCoAs, so this is a PCoA derived analysis.

 Figure 3: Stationary 3D figures are a really bad idea. They are incredibly difficult to interpret properly without being able to manipulate the angle of the 3D space.

We now provide a 2D figure for improved interpretability in print. We have also moved some panels to supplemental. We have also made the 3D figure files available for interactive manipulation of the aesthetics desired by a given reader.

 Line 273-275: Unsurprisingly. One would expect a larger difference at the lower taxonomic levels.

We agree. However this has not been explicitly shown for museum specimens thus why it is mentioned here.

 Line 275: “comp16ared” should be “compared”

Revised.

 Lines 283-284: You are referring to Figure 6 before referring to Figure 5. This should be sequential.

Revised with the updated figures.

 Lines 287-289: More importantly than being an outlier, shotgun (WoL) and metabarcoding (GG) agreeing on the rickettsia abundance while shotgun (Rep200) disagrees is an interesting finding. It makes me question the validity of the Rep200 analysis, since this is a very obvious result that should have been recovered too. I hope to find a discussion of this later in the Discussion part. Although, rickettsiales could also potentially mean another case of mitochondrial contamination.

We have added discussion on this point in the Discussion section. Mitochondrial filtering was completed, but taxonomic classification for intracellular bacteria continues to be controversial and this may reflect different naming conventions between databases.

 Figures 4, 5, 6: It would be easier if these three figures were maybe combined into one multipanel figure, and also the order of the X axis being made the same in both. As it’s standing, it is very difficult to compare the three methods. (also, the dimensions of the figures differ)

We agree and have updated the figures as suggested by reviewers.

 Heatmaps: not sure if they add anything to the argument. (To the point where the only mention of figure 7 is with figures 4-6 in a single sentence.)

We believe the heatmaps provide a different visual style that some readers may find useful to convey the point being made. We have therefore kept this figure (albeit revised) in the primary article. We have included additional text in the manuscript referencing these figures.

 Lines 317-319: A comparison is not as straight forward as the authors make it seem. Especially looking at the diversity measures, I would not be inclined to compare between shotgun (WoL and 16S metabarcoding results. The authors state “at times” but when can they really be sure the time is right? Yes, the bar graph shows strong similarity (as far as I can see), but I don’t think this is enough cause.

We agree that 16S and shotgun comparisons are non-trivial, especially when considering representation of abundances in diversity metrics. This is why we used Jaccard distance matrices for our Procrustes analysis and ACE for alpha diversity comparisons, as discussed.

 Lines 333-335: It think it’s still possible to compare. It’s not a bias free comparison, but that’s not the point. You are not trying to compare Treatment A done with metagenomics with Treatment B done with 16S, I agree that would be impossible due to the non-trivial problems. However, comparing how metagenomics and 16S differ in comparing Treatment A and B is a valid analysis, that is independent of these abundance-linked issues. These issues are part of the nature of either sequencing method and are integral to the comparison.

We agree that our comparisons are valid, as the material being compared are of the same treatments (i.e. methodological comparison).

 Supplementary Table 2: since you used DADA2, it would be interesting to see the full count table DADA2 produces for read filtering quality control, not just raw counts.

This has been included as an additional column in the supplemental file.

---

## [Decision Letter · Decision Letter 1]

1 Sep 2023

Shotgun Metagenomics Captures More Microbial Diversity than Targeted 16S rRNA Gene Sequencing for Field Specimens and Preserved Museum Specimens

PONE-D-23-13742R1

Dear Dr. Madison,

We’re pleased to inform you that your manuscript has been judged scientifically suitable for publication and will be formally accepted for publication once it meets all outstanding technical requirements.

Kind regards,

Ruslan Kalendar

Academic Editor

PLOS ONE

Reviewers' comments:

Reviewer's Responses to Questions

**Comments to the Author**

Reviewer #2: 

The author's have addressed all of my points adequately.

A minor comment, but I wonder how useful figures 2 and 3 are for this manuscript. But they don't take away from the message. They just don't add much.

---

## [Editor Report · Acceptance letter]

10 Sep 2023

PONE-D-23-13742R1 

Shotgun Metagenomics Captures More Microbial Diversity than Targeted 16S rRNA Gene Sequencing for Field Specimens and Preserved Museum Specimens 

Dear Dr. Madison:

I'm pleased to inform you that your manuscript has been deemed suitable for publication in PLOS ONE. Congratulations! Your manuscript is now with our production department. 

Kind regards, 

on behalf of

Professor Ruslan Kalendar 

Academic Editor

PLOS ONE